# The Relationship between Compulsive Exercise, Self-Esteem, Body Image and Body Satisfaction in Women: A Cross-Sectional Study

**DOI:** 10.3390/ijerph19031857

**Published:** 2022-02-07

**Authors:** Juncal Ruiz-Turrero, Karlijn Massar, Dominika Kwasnicka, Gill A. Ten Hoor

**Affiliations:** 1Department of Work and Social Psychology, Maastricht University, P.O. Box 616, 6200MD Maastricht, The Netherlands; juncal.ruiz@hotmail.com (J.R.-T.); karlijn.massar@maastrichtuniversity.nl (K.M.); 2Faculty of Psychology, SWPS University of Social Sciences and Humanities, Aleksandra Ostrowskiego 30b, 53-238 Wrocław, Poland; dkwasnicka@swps.edu.pl; 3NHMRC CRE in Digital Technology to Transform Chronic Disease Outcomes, School of Population and Global Health, University of Melbourne, 333 Exhibition Street, Melbourne 3000, Australia

**Keywords:** self-esteem, body satisfaction, body image, compulsive exercise, physical activity

## Abstract

Purpose: In this study, we aimed to test the relationship between compulsive exercise and self-esteem, body image, and body satisfaction as potential predictors of eating disorders. Methods: Self-report measures of compulsive exercise beliefs and behaviors, self-esteem, body image, and body satisfaction, were completed by 120 female participants through an online questionnaire. Factor analyses with varimax rotation were performed to create exercise-frequency groups. ANOVA’s were performed on Body Mass Index (weight adjusted for height squared, BMI), current minus ideal weight, self-esteem, body image, and body satisfaction to determine if there were differences depending on these exercise groups. Results: Factor analysis revealed two factors for compulsive exercise beliefs and behaviors: (1) Exercise Fixation, and (2) Exercise Frequency and Commitment. Based on those factors, participants were subdivided into four clusters: (1) pathological obligatory exercisers, (2) exercise fixators, (3) committed exercisers, and (4) non-exercisers. No differences were found between these groups for BMI or current minus ideal weight. Pathological obligatory exercisers and committed exercisers spend significantly more hours on exercise weekly compared to exercise fixators or non-exercisers. No differences between pathological obligatory exercisers, exercise fixators, and non-exercisers were found on self-esteem or body satisfaction, where the committed exercisers scored significantly better. Both committed exercisers and non-exercisers scored significantly better on body image flexibility compared to pathological obligatory exercisers and exercise fixators. Conclusion: Compulsive exercise has both an exercise volume and an emotional component. The level of compulsive exercise is linked to one’s self-esteem, body image, and body satisfaction and those factors could be a target for future interventions.

## 1. Introduction

The benefits of physical exercise for individuals’ mental and physical health are well documented. Specifically, exercise is known to decrease depression and anxiety [1,2], to increase energy levels in both healthy individuals and those who suffer from various medical conditions [3,4,5], to reduce the risk of chronic disease [6,7], and to increase brain health and memory [8]. However, for some individuals, there is a slippery slope from healthy and enjoyable exercise to *compulsive* exercise, which can cause interference with one’s daily activities and is often undertaken to a level that is harmful [9]. Importantly, compulsive exercise is a common symptom of several eating disorders, predominantly of bulimia and (the purging form) of anorexia nervosa [10,11].

The American National Eating Disorder Association [12] has defined compulsive exercise as one or more of the following: “(1) Physical exercise that significantly interferes with important activities, occurs at inappropriate times or inappropriate settings, or when the individual continues to exercise despite injury or other medical complications; (2) Intense anxiety, depression and/or distress if unable to exercise; (3) Discomfort with rest or inactivity; (4) Exercise is used to manage emotions; (5) Exercise is used as a means of purging; (5) Exercise is used as permission to eat; (6) Exercise that is secretive or hidden; and (7) Feeling as though you are not good enough, fast enough or not pushing hard enough during a period of exercise.” Research has established that for compulsive exercisers, not being able or allowed to exercise causes increased negative effects (including distress, guilt, anxiety; e.g., Yates, 1991) [13].

Whether physical exercise has positive or negative psychological effects and is mainly determined by the motivations behind it. Physical exercise is often performed not solely for health-related reasons but for appearance-related reasons, i.e., to manage shape, weight, and muscularity [14,15]. Indeed, the risk for compulsive exercise is larger among those who exercise for weight and shape control compared to individuals who exercise for health promotion reasons [16]. Women especially engage in exercise motivated by weight and shape control which in turn was associated with lower body satisfaction [15]. Similarly, other researchers also provided evidence that exercise for appearance-based reasons was associated with lower body satisfaction, lower body esteem, and lower self-esteem [17]. Indeed, compensatory and compulsive features of exercising were better predictors of disordered eating and eating disorder diagnoses than exercise that was excessive in quantity [18].

Eighty-three percent of women in the United States report that they are not happy with their bodies [19], and there are clear indications that body dissatisfaction is associated with both disordered eating [20] and excessive exercising [21]. Moreover, there is a direct link between body (dis)satisfaction and general subjective well-being [22], as well as between body (dis)satisfaction and self-esteem [23,24]. Body image and physical activity have a bi-directional relationship—whereas a positive body image is associated with increased physical activity (and a negative body image with decreased engagement in physical activity and sport), engagement in physical activity and sport also affect one’s body image [20,25]. It is less clear, however, how body satisfaction, body image, and self-esteem are associated with particular *subtypes* of exercisers.

In sum, the literature indicates that what distinguishes compulsive exercisers from healthy exercisers is not the frequency or quantity of the exercise, but the underlying motivations, the emotional attributions, and the psychological meaning that is attached to (not) exercising [18,22,26]. Therefore, in the current study, we focus on identifying these motives in women (the group most at risk for developing pathological eating habits). Knowledge of such psychosocial motivations underlying (compulsive) exercise can facilitate timely identification of at-risk individuals. Further, to be able to create effective, tailored health promotion interventions aimed at preventing or reducing disordered eating, it is thus useful to define ‘exercise profiles’ and investigate the underlying psychological variables distinguishing different types of exercisers. Therefore, in the current study, following and extending the study by [26] who performed a similar study on the relationship between (profiles of) excessive exercise and disordered eating behaviors, we investigated exercise typologies in a sample of women and assessed the association of (compulsive) exercise with BMI, ideal weight, self-esteem, body satisfaction, and body image. Although this research is exploratory, we do expect that for women that score higher on the emotional (‘fixation’ or ‘obligation’) component of exercising, scores on body image, body satisfaction, and self-esteem will be lower than for women who exercise for other reasons.

## 2. Method

All study materials, data, and syntax can be found in Appendix A on https://osf.io/u38gh/?view_only=24124371c9b9424da388ab8c46097faa (accessed on 21 December 2021).

### 2.1. Participants and Procedure

Participants over 16 years of age (The ethics committee at Maastricht University allows participants to consent when they are 16 years or older without parental co-consent) were recruited through (1) the university undergraduate psychology participant pool from Maastricht University, the Netherlands and (2) through additional advertisements on social media with a direct link to the Qualtrics-hosted survey. In total, 216 participants provided responses. After removing incomplete responses (*n* = 51) and male participants (*n* = 45), the final dataset consisted of *n* = 120 females (16–54 years old; Mean age = 22, *SD* = 7). Study participants were mostly German (*n* = 46, 53.5%), Spanish (*n* = 37; 30.8%), or Dutch (*n* = 18; 15%). Aside from minimum age and gender, we did not specify inclusion or exclusion criteria. The entire survey was in English, as this is the preferred language of communication at our university [27].

After reading a participant information sheet and providing informed consent, participants proceeded to (1) demographics questions, (2) the Obligatory Exercise Questionnaire, (3) the Rosenberg Self-Esteem Scale, and (4) the Body Satisfaction Scale, and (5) the Body Image—Acceptance and Action Questionnaire (see also measures). At the end of the survey, participants were thanked and debriefed. The questionnaire had a total of 82 items. It took participants an average of 16 (±13) minutes to complete (three extremes ranging from 3.5 h to two days were not considered in this average—the rest of their data were included as they did not show any abnormalities). Students received participation points; all other participants were not remunerated. This study was approved by the Ethics Review Committee of the School of Psychology and Neuroscience at Maastricht University, The Netherlands (ERCPN number: 188_10_02_2018_S20).

### 2.2. Measures

#### 2.2.1. Demographics

Participants were asked to report their nationality, age, gender, weight, ideal weight, and height. In terms of the participants’ “ideal weight”, they were asked to estimate if participants wanted to gain or lose weight, and was also used to calculate their ideal BMI and compare it to their current BMI. Additionally, participants’ frequency of physical activity was measured by asking them to self-report the number of hours per week (on average) that they exercised.

#### 2.2.2. Obligatory Exercise Questionnaire (OEQ)

The OEQ [28] measures the extent to which individuals have obligatory (compulsory) exercise beliefs and behaviors. Respondents rated 20 items (e.g., “When I don’t exercise, I feel guilty” from 1: Never—4: Always. If needed, items were recoded to make sure that a higher score indicated more obligatory exercise beliefs and behaviors. The internal consistency of this questionnaire was high (Cronbach’s alpha is .86, comparable with what was found by other researchers [28], α = .96).

#### 2.2.3. The Rosenberg Self-Esteem Scale (RSES)

The RSES [29] measures individuals’ self-esteem. Respondents rate 10 items (e.g., “I certainly feel useless at times”) from 1: Strongly agree to 4: Strongly disagree. After recoding some items, a sum score was calculated (ranging from 10 to 40 points), where a higher score indicated lower self-esteem.). Internal consistency of this measure was high (α = .85 and .88; [29]. In the present study, the Cronbach’s alpha was .88.

#### 2.2.4. The Body Satisfaction Scale (BSS)

The BSS [30] measures participants’ degree of satisfaction with their different body parts such as teeth, nose, or tummy. Respondents rated 16 body parts from 1: Very satisfied to 7: Very unsatisfied. A sum score was calculated (range 16 to 112 points) where a higher score indicated lower body satisfaction. Internal consistency of this measure was high (α = .97 by [30] and α = .86 in this study).

#### 2.2.5. The Body Image—Acceptance and Action Questionnaire (BI-AAQ)

The BI-AAQ [31] measures the extent to which individuals exhibit an accepting posture towards negative thoughts and feelings about their body shape and/or weight. Respondents rated 12 items (e.g., “I care too much about my weight and body shape”) from 1: Never true to 7: Always true. After some items were recoded, a final sum score was calculated (ranging from 29 to 203 points), where a higher score indicated greater body image flexibility. Internal consistency of this measure was high (α = .93 in both this study as the study by other authors [31].

### 2.3. Statistical Analyses

The present study followed the same line of statistical analyses as previous one on this topic [21]. First, a principal component factor analysis with varimax rotation was performed to reduce the number of OEQ items. Then, participants were clustered into different groups depending on their OEQ subscale scores. A multivariate analysis of variance was performed on weight, ideal weight, height, BMI, ideal BMI, the number of hours per week on physical activity. Another multivariate analysis was performed on the scores on self-esteem, body image, and body satisfaction scales to measure if there were statistically significant differences depending on the cluster to which the participants belonged.

## 3. Results

As most participants were young adult women, we ran the analyses with all women (age range 16–54) but also with all women within 2 SD of the mean age (age range 16–30, with M = 20.65 and SD = 2.43). Since the outcomes did not change when the older women were excluded, we decided to report the results for the total sample here. BMI was calculated using the participants’ self-report measures of their weight and height. Results showed that 9.3% (*n* = 11) of participants were underweight, 66.9% (*n* = 79) had a healthy weight, 16.1% (*n* = 19) were overweight, and 7.6% (*n* = 9) were obese (two participants did not indicate their height or weight). The mean BMI was 22.52 (SD = 4.44). On average, the ideal BMI was 2 points lower (M = 20.86, SD = 2.88), and the ideal weight was 4.4 kg’s (SD = 6.47 kg) lower than their current weight (Range −40 to +10 kg’s).

### 3.1. Principal Components Factor Analyses

A series of principal components factor analyses with varimax rotation were performed in order to reduce the number of OEQ items. Following each analysis, items that did not load at least .5 on one factor were eliminated. After three analyses, nine items were eliminated, with the remaining 11 items loading on three factors that explained the 62.5% of the variance in the item set. However, the third factor consisted of just one item (“My best friend likes to exercise”). This factor had an Eigenvalue of 1.078 and was not used for further analyses. The remaining 10 items were subdivided by their designated factors and factor loadings (see Table 1).

### 3.2. Creating Groups

Participants were clustered into four different groups using their OEQ subscale scores. Table 2 contains standardized descriptive statistics for the four cluster-analysis-derived variables. The members of Group 1 (*n* = 29) score higher than average on both factors and are identified as ‘pathological obligatory exercisers’. Those in Group 2, the ‘exercise fixators’ (*n* = 25) score higher than average on exercise fixation but lower than average on exercise frequency and commitment. In Group 3, the ‘committed exercisers’ score higher than average on exercise frequency and commitment, but lower than average on exercise fixation (*n* = 26). Lastly, members of group 4, the ‘non-exercisers’ (*n* = 40) are characterized by a lower than average score on both factors.

### 3.3. Differences between Groups

Multivariate analyses of variance (MANOVA) were performed on measures of BMI, and a difference score between current weight and ideal weight (‘delta weight’). There was no significant effect on group level on the BMI or difference between weight and ideal weight, λ = .94, *F* (6, 228) = 1.25, *p* = .28 (non-significant univariate outcomes are reported in Table 3). A follow-up analysis using univariate ANOVA on the number of hours per week that they spend performing physical activity showed significant differences between groups (*F* (3, 115) = 1.10, *p =* .35) (see Table 3 for post-hoc results, where different letters behind the means mean a significant difference between groups). Lastly, differences in self-esteem, body satisfaction, and body image were examined in a MANOVA. Wilks’ statistics show a significant effect of group on self-esteem, body satisfaction, and body image. λ = .75, *F* (9, 277.60) = 3.88, *p* <.001. Separate follow up univariate ANOVA’s on the outcome variables revealed significant between-group effects on self-esteem (*F* (3, 116) = 4.84, *p =* .003), body satisfaction (*F* (3, 116) = 3.46, *p =* .02), and body image (*F* (3, 116) = 10.24, *p* <.001). For example, Pathological Obligatory Exercisers and Exercise Fixators scored significantly lower on body image and body satisfaction than committed and non-exercisers. All mean scores, SDs, and test results (including simple effects) can be found in Table 3. 

## 4. Discussion

The main purpose of this study was to investigate exercise typologies in a sample of women and evaluate the association of (compulsive) exercise with BMI, ideal weight, self-esteem, body satisfaction, and body image as potential correlates of eating disorders. To accomplish this, and replicating [26], participants were divided into four groups, which were determined by their scores on Exercise Fixation and Exercise Frequency and Commitment scale.

Results showed significantly lower levels of self-esteem, body satisfaction and body image for the groups that rated higher on Exercise Fixation: The Pathological Obligatory Exercisers and the Exercise Fixators. These results are in line with previous findings [26], demonstrating that Exercise Fixation is a risk factor for developing an eating disorder pathology [32].

On the other hand, the Committed Exercisers (high on exercise frequency but low on exercise fixation) consistently showed significantly higher levels of self-esteem, body satisfaction and body image. The non-exercisers showed low self-esteem, average body satisfaction and high levels of body image. These results are also consistent with previous studies [33] demonstrating that female athletes feel less social physique anxiety and report higher levels of self-esteem and body image-satisfaction. In line with other studies, showing that athletes’ high physical activity levels and bodies may more closely resemble the current aesthetic ideal of a thin/lean and fit physique for females [34].

### Strengths and Limits

Several implications can be drawn from the present study. First, these findings demonstrated that compulsive exercise is not unidimensional, and thus researchers investigating the link between compulsive exercise and other pathologies, should apply multidimensional measures of compulsive exercise, such as the one applied in this study. Morgan (1979) was the first to suggested that compulsive exercise is not just a matter of physical exercise volume, but instead, it also has an emotional component [35]. This study showed support for this suggestion, as it was found that it was not exercise frequency and commitment that predicted different outcomes for self-esteem, body satisfaction and body image, but instead, it was the exercise fixation that predicted the changes in these measures between the four cluster-analysis-derived groups.

The current findings also expand those of previous studies by explaining the link that has been found between (compulsive) exercise and eating pathologies. Ackard and colleagues (2002) found that the Exercise Fixation factor showed the highest correlations with most Eating Disorder Inventory (EDI) subscales [26], which suggested a link between this factor and eating disorder traits and behaviors. In this study, we aimed to explore this link further and found the same factor to predict the lowest levels of self-esteem, body image and body satisfaction. These three measures may help to explain the relationship between compulsive exercise and eating disorders. It is possible that the relationship between compulsive exercise and eating disorders is mediated or moderated by the levels of self-esteem, body image and body satisfaction, but further research is needed to answer the aforementioned research questions.

This study has some limitations, and therefore results should be interpreted cautiously. First, this study was merely cross-sectional, and responses were collected using an online questionnaire. This has led to a) the possibility that the collected data on, e.g., self-reported weight or height were potentially not reported correctly (even though these variables were relatively normally distributed), and b) that we cannot conclude that participants’ low levels of self-esteem, body image, and body satisfaction led to higher scores in exercise fixation or vice versa. Second, data collection was done by a University student (1st author), and using university communication channels. Therefore, most participants were students, and therefore the results could not be generalized to the general population. Future studies may add objective measures of, e.g., BMI to ensure that self-reported data matches objectively measured data. Additionally, the sample size of this study was relatively small compared to previous studies. For example, other research included 586 participants [26], having clusters ranging from 52 to 180 participants in each of them. In our study, we only analyzed 120 female participants, and our clusters were much smaller, but the distribution of participants across the clusters was equal. We specifically lacked sufficient male participants in our sample, precluding the possibility of making comparisons with the female participants. For interested readers, the additional data for men is made available at https://osf.io/u38gh/?view_only=24124371c9b9424da388ab8c46097faa (accessed on 21 December 2021). Future researchers should collect comparable sample sizes of male and female participants to study the reported patterns across genders. 

## 5. Conclusions

In conclusion, this study suggests that the compulsive exercise measure is multidimensional. Compulsive exercise has several components, including exercise volume, an emotional component, and is linked to one’s self-esteem, body image, and body satisfaction, and could therefore potentially be a target for future interventions (see also [36]. We demonstrated that, based on their scores in these measures, participants can and should be divided into different groups with differential relationships to other variables. Interestingly, our findings show that the emotional element of exercise has the most aversive effects on individuals’ self-esteem, body image, and body satisfaction, and not compulsive exercise scores as a whole. Next to cross-sectional studies (e.g., [37]), future research should involve experimental studies where measures are manipulated to investigate the causal direction of the effect [36]. For example, a study that manipulates participants’ levels of self-esteem and compares scores on the OEQ scale between conditions. Longitudinal studies can also provide valuable knowledge on the link between compulsive exercise and eating disorders by determining clear patterns over time. 

### 5.1. What Is Already Known on This Subject? 

Physical exercise is healthy, up to a certain level where this becomes *compulsive* exercise. There is a bi-directional link between body image and (excessive) exercise. The current literature indicates that it is not the frequency or quantity of exercise, but the underlying motivations, the emotional attributions, and the psychological meaning that is attached to (not) exercising that distinguishes compulsive exercising from healthy exercise.

### 5.2. What Does This Study Add?

This study adds information on how BMI, ideal weight, body satisfaction, body image and self-esteem are associated with particular subtypes of exercise. 

## Figures and Tables

**Table 1 ijerph-19-01857-t001:** Factor Loadings of 10 Remaining OEQ Items.

Factor	OEQ Item ^a^	Factor Loading	Cronbach’s α
Factor 1: Exercise Fixation	4. When I don’t exercise, I feel guilty	.80	.83
	13. When I miss a scheduled exercise session, I may feel tense, irritable or depressed	.76	
	7. When I miss an exercise session, I feel concerned about my body possibly getting out of shape	.74	
	14. Sometimes, I find that my mind wanders to thoughts about exercising	.71	
	12. If I feel I have overeaten, I will try to make up for it by increasing the amount of exercise	.71	
	15. I have had daydreams about exercising	.60	
Factor 2: Exercise frequency andcommitment	3. I exercise more than three days per week	.82	.80
	1. I engage in physical exercise on a daily basis	.82	
	2. I engage in one/more of the following forms of exercise: walking, jogging/running or weight-lighting	.76	
	10. I may miss a day of exercise for no reason	.69	

^a^ Original OEQ item numbers.

**Table 2 ijerph-19-01857-t002:** Means, Standard Deviations, and Group Size (*n*) of Four Clusters on OEQ Subscales.

	OEQ Subscales	
Group (*n*)	Exercise Fixation	Exercise Frequency and Commitment
Pathological obligatory exercisers (*n* = 29)	.96 (.82)	1.00 (.57)
Exercise fixators (*n* = 25)	.78 (.67)	−.55 (.47)
Committed exercisers (*n* = 26)	−.61 (.42)	.76 (.38)
Non-exercisers (*n* = 40)	−.79 (.45)	−.88 (.69)

Standardized values.

**Table 3 ijerph-19-01857-t003:** Mean scores, Standard Deviations (SDs) and test results (including simple effects).

	Group				Univariate ANOVA’s
Outcome	1. Pathological Obligatory Exercisers	2. Exercise Fixators	3. Committed Exercisers	4. Non-Exercisers	
BMI *	23.61 (4.69) ^a^	22.55 (3.59) ^a^	22.04 (5.54) ^a^	21.75 (3.62) ^a^	*F* (3, 115) = 1,10, *p* = .35, partial *ƞ*^2^ = .03
Delta weight *	−4.76 (3.78) ^a^	−5.72 (5.97) ^a^	−3.96 (8.31) ^a^	−3.58 (7.08) ^a^	*F* (3, 115) = 0.63, *p* = .60, partial *ƞ*^2^ = .02
Exercise hours	5.64 (4.38) ^a^	2.86 (0.48) ^b^	6.10 (0.46) ^a^	2.45 (0.37) ^b^	*F* (3, 116) = 19.14, *p* < .001, partial *ƞ*^2^ = .33
Self-esteem	23.44 (4.68) ^a^	22.64 (5.11) ^a^	18.53 (4.55) ^b^	21.40 (5.56) ^a^	*F* (3, 116) = 4.84, *p* = .003, partial *ƞ*^2^ = .11
Body satisfaction	50.51 (14.35) ^a^	49.60 (13.05) ^a^	39.38 (12.55) ^b^	45.83 (15.19) ^a,b^	*F* (3, 116) = 4.84, *p* = .003, partial *ƞ*^2^ = .08
Body image	48.97 (14.44) ^a^	55.36 (13.34) ^a^	65.92 (9.29) ^b^	63.40 (14.58) ^b^	*F* (3, 116) = 10.24, *p* < .001, partial *ƞ*^2^ = .21

* Weight—ideal weight. For BMI and weight, there was one participant missing who did not enter weight correctly. ^a,b^ same letters mean no significant difference between groups, different letters mean significant differences between groups.

## Data Availability

All materials, data and syntax are available via [https://osf.io/u38gh/?view_only=24124371c9b9424da388ab8c46097faa] (accessed on 21 December 2021).

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
