# Peer review of "The Relationship between Compulsive Exercise, Self-Esteem, Body Image and Body Satisfaction in Women: A Cross-Sectional Study"

_ijerph, 2022, doi:10.3390/ijerph19031857_

Round 1

Reviewer 1 Report

The analysis of the relationship between self-esteem, satisfaction with the body and compulsive exercise is old, where in a brief survey we could find studies dating from 1969, where the interest was to find the term physical fitness and its relationship with healthy behaviors. The study is interesting, in a basic correlational analysis in an online questionnaire composed of several instruments, but it demonstrates some weaknesses and lacks some factors to clarify the reader. My assessment will be global on some points and specific on others.
Why use only part of the data and not all the data?
Abbreviations that were not previously defined by readers appear in the abstract and throughout the text (eg BMI). It is important to define what they mean before using them, readers with less knowledge on this topic may not understand what they mean.
In the introduction, aren't there more recent studies talking about the indicated topics? Due to having references over 30 years old and others over 20 years old.
The methodology is well described with the inclusion of ethics committee approval, but I ask why not using the male gender? Since I made this collection.
The age distance (16-54 years) can influence their results, as behavior and body image change with advancing age. Since your average is 22 years.
Another aspect due to the online application is the non-confirmation of the collection data, believing that the participants gave the correct information, for example, their weight, height, etc., which translates into a weakness in their study.

it would become more robust if it managed to balance the number of male and female participants.
Something that worried me was that the data presented do not have inclusion and exclusion criteria.
I ask why use the ANOVA test and not the MANOVA test, which would give more robust results? This indication of MANOVA only appears in the results.
In the results, the description of the groups must be included in the methodology and may or may not be reinforced in the results.
The presentation of values ​​in parentheses must be homogeneous, e.g. (n=25) and then another appears with (N=26), review the entire text. Here, consistency in presentation is also required.
After a good discussion, expect a good descriptive conclusion, not just a paragraph.
This topic has already been addressed a few times in the journal, where not presenting references does not represent the real interest in the works published by the journal. Another factor is the fact that a reference in your bibliography only shows the title, without indicating the place of publication.
I realize how difficult it is to carry out such a study, but my recommendation as it stands is to reject it.

Author Response

We attach our entire response here. We thank the reviewer for their helpful feedback. 

Reviewer 2 Report

The manuscript is in the scope of the Journal. The idea of the research can be interesting for some readers, the manuscript is written clearly and comprehensively. However, although some significant concerns may be successfully addressed, I think that the conceptual value of the manuscript is disputable and it does not have significance to move the field forward meaningfully. The research question is not original, relevant, and well-defined and the study results do not provide important data to the field. 

Name of the manuscript: 

  • The name of the manuscript lack information on the sample. A sample consisted of young women and the reference for this information should be included in the name.

Introduction:

  • Lines 79-81: the authors argue that “what distinguishes compulsive exercisers from healthy exercisers is not the frequency or quantity of the exercise, but the underlying motivations, the emotional attributions, and the psychological meaning that is attached to (not) exercising”. Such argument is based on the empirical data, however, the authors do not pursue their argumentation and instead study the association of exercise with BMI, ideal weight, self-esteem, body satisfaction, and body image. From the reader’s point of view, such a decision is unclear – why did the authors choose these variables, not the motivation of the exercise, etc.? This part of the Introduction needs more clarification as well as the decision to study young women.
  • It would be helpful if the authors provide the main research question(s) (hypothesis) and its justification.
  • A substantial part of the introduction is based on the old scientific literature even though there is a large body of new (within the last 5 years) data. Thus it is recommended to focus on the newest literate and include the older references only if this is necessary to convey the authors’ arguments.

Method:

  • Line 98: the description of the sample reveals a very wide range of participants' age (16-54 years old) although the mean age (22) shows that most women were young. At the end of the Introduction, it is written that “young women” will be studied, however, the age of 54 can not be defined as “young”. It is recommended to remove the data of participants whose age is more than one SD. Additionally, it is stated that “Participants over 18 years of age were recruited” (line 94), however in line 98 there is information that the youngest participant was 16 years old. Such discrepancies must be eliminated.
  • There is no information on informed consent – what information was provided? Besides, in most countries, a person of 16 years is defined as a minor who needs parents’ informed consent for participation in studies. Thus, information must be provided on who gave informed consent in the case of participants under 18 years old.
  • Lines 104-105: “It took participants an average of 16 (± 13) minutes to complete (three extremes – ranging from 3.5h to 2 days - were not considered in this average)”. What was done with that data (was it included in the analysis or not)? 
  • Line 112: what was the reasoning behind including a question about participants’ nationality? The reasoning is important, because if the language of the survey was English (it is not written but should be if respondents with different nationalities participated in the survey), how participants’ understanding of survey questions was controlled if their mother-tongue was not English (the Germans, the Spanish, the Dutch)?

Results

  • As the research questions were not presented the logic of some data analysis is questionable. For example, why “multivariate analyses of variance were performed on measures of BMI, and a difference score between current weight and ideal weight (‘delta weight’). <…> A univariate ANOVA on the number of hours per week that they spend performing physical activity showed significant differences between groups” was needed?
  • The post-hock analysis was not performed and there is no possibility to say what group differed from what. It is not enough to say that groups differ, more important is knowledge on specific significant differences between concrete groups. I would highly recommend conducting a post-hock analysis to obtain these results. 

Discussion 

  • Lines 203-205: “The main purpose of this study was to investigate exercise typologies in a sample of 203 young women and evaluate the association of (compulsive) exercise with BMI, ideal 204 weight, self-esteem, body satisfaction, and body image, as potential predictors of eating 205 disorders”. The conducted statistical analysis does not allow making conclusions about predictors. 
  • Lines 209-210: “Results showed significantly lower levels of self-esteem, body-satisfaction and body-209 image for the groups that rated higher on Exercise Fixation: The Pathological Obligatory 210 Exercisers and the Exercise Fixators.” This statement is misleading, as the post-hock analysis was not conducted. It may be that group 1 differs from group 3, but not from groups 2 and 4, etc. The visual difference of scale means does not mean a significant difference between groups. Because the discussion greatly relies on the colligated results its usefulness is very questionable.

Author Response

(The authors gave the same response as above.)

Reviewer 3 Report

The relationship between compulsive exercise, self-esteem, body-image and body-satisfaction is very important topic. Currently, a large population of young people suffer from low self-esteem and seek to improve their physical appearance in relation to the current "model of beauty".

The presented article requires some corrections.

Please explain why authors decided not to include 45 males in statistical analysis? I suggest adding to the topic of the work the phrase: among young women or preliminary research/if authors plan to continue the study with men included.

The discussion of the results is too short and superficial. Sentences containing information that the obtained results are similar, the same or different from other scientific reports are not sufficient. It is worth including a reflection on this subject, a description of the mechanism. What this dependence may result from, according to the authors of the article? The part of the introduction citing research by other authors may be included in the discussion of the results of the research.

Author Response

(The authors gave the same response as above.)

Reviewer 4 Report

These authors have certainly presented important and interesting  information about the motivation and significance of various factors involved in attaining and maintaining compulsive exercise behavior.  Nevertheless, there are still a few overall concerns about this manuscript.  While the experimental design is quite appropriate for getting at the attitudes and beliefs associated with each individual’s self-perception of his/her own body, there is for practical reasons a problem in that they are focusing on self-perception but have no actual measures of the body’s form and function.  In the reviewer’s experience this information is obviously needed to determine the reliability and validity of these self-reports, the discrepancy between self-report and the actual body morphology and function providing important diagnostic information, e.g., anorexia.  Given logistic problems in obtaining such data, perhaps a small subset of selected willing volunteers (recognizing the “volunteer error” problem) could provide some actual measures for determining a correction factor based on differences between actual and self-report measures.

Although certainly impressive, the very sophisticated statistical analyses and results presented in this manuscript could be expressed in more user-friendly terms to impart these results and other information.

Finally, the wording of some of the questions must be made more precise by not adding qualifying words such as “possibly” to the otherwise straightforward questions; for example, in Table 1, Factor 1: Exercise Fixation, Question #7. “When I miss an exercise session, I feel concerned about my body possibly getting out of shape.”   Perhaps a more standard Likert or similar scale could avoid uncertainty about the meaning or magnitude of the response.

Author Response

(The authors gave the same response as above.)

Round 2

Reviewer 1 Report

Dear authors,

I appreciate the care and acceptance of the suggestions in order to enhance the decisions and writing of the article.

As I do not have access to the global results that will be made available online with the men's data, I continue with this question of including them in your study, which would bring greater robustness to the results.

Thank you for clarifying some doubts.
Inclusion and exclusion criteria can be defined in the text.
e.g. Women; ages between x and y years; belonging to university z; with regular practice of physical exercise.
Excluding, for example, failure to complete the questionnaire, musculoskeletal injuries in the last 3 months, etc.

Comparative studies have their value but they bring little reliability when carried out by autoresponder to variables as important as anthropometric and body data, which are a structuring part of your investigation.

I appreciate the improvements presented but my recommendation is of approval with the slight revision of these details and concerns that I continue to present.

Reviewer 2 Report

The authors made some corrections; however, these corrections do not add conceptual value to the manuscript.

I still recommend substantiating the scientific problem (why it is important to study variables the authors researched?). Additionally, some explanations the authors provided for a reviewer must be included in the manuscript (e.g. information regarding participants' consent age in the country). And some explanations do not add a better understanding thus must be reconsidered and corresponding corrections must be made in the manuscript (e.g. information regarding post-hock results. The note under table 3 is incomprehensible and readers will not understand the results regarding between-group differences).
